# Automated Classification of Ultrasonic Signal via a Convolutional Neural Network

Yakun Shi [1], Wanli Xu [1], Jun Zhang [1] and Xiaohong Li [2,*]

[1] School of Power and Mechanical Engineering, Wuhan University, Wuhan 430072, China; shiyakunwhu@163.com (Y.S.); wanlixu@whu.edu.cn (W.X.); zhangjun2010@whu.edu.cn (J.Z.)
[2] School of Materials and Engineering, Southeast University, Nanjing 211189, China
* Correspondence: xiaohongli@whu.edu.cn

**Abstract:** Ultrasonic signal classification in nondestructive testing is of great significance for the detection of defects. The current methods have mainly utilized low-level handcrafted features based on traditional signal processing approaches, such as the Fourier transform, wavelet transform and the like, to interpret the information carried by signals for classification. This paper proposes an automatic classification method via a convolutional neural network (CNN) which can automatically extract features from raw data to classify ultrasonic signals collected of a circumferential weld composed of austenitic and martensitic stainless steel with internal slots. Experiments demonstrate that our method outperforms the traditional classifier with manually extracted features, achieving an accuracy rate of classification up to 0.982. Furthermore, we visualize the shape, location and orientation of defects with a C-scan imaging process based on classification results, validating the effectiveness of the results.

**Keywords:** ultrasonic signal; automated classification; features; signal processing; convolutional neural network





## 1. Introduction

Ultrasonic testing is a versatile nondestructive testing (NDT) technique for the quality assessment and identification of defects inside a wide variety of materials, including metals, plastics, ceramics and composites since ultrasonic waves can propagate in these materials as a form of mechanical vibration [1]. Ultrasonic inspection has many excellent performances [2], such as its proficiency in flaw location and size specifications and its high sensitivity to damages. The most commonly used ultrasonic testing approach is the pulse-echo method for its simplicity, accuracy and efficiency. In this method, a piezoelectric transducer is used to generate ultrasonic waves propagating through the inspection object. Once a defect is encountered, parts of waves will be reflected and return back to the transducer. Such waves will be converted to electrical signals, also called A-scan signals, containing information about the location, type, size and orientation of the defect [3]. At present, the interpretation of ultrasonic signals is usually accomplished manually. The results are prone to be influenced by human factors heavily reliant on inspection personnel's experience and knowledge. In addition, the process will be laborious and time-consuming as the collected data drastically increases. An automatic classification system interpreting ultrasonic signals accurately and consistently is becoming the urgent need of industries for minimizing errors caused by inspectors' subjectivity [4].

In recent decades, the significant progress of artificial intelligence techniques makes it possible to automatically classify ultrasonic signals. Such techniques contain ultrasonic pattern recognition, neural networks, etc. Neural networks for ultrasonic signal classification have attracted much attention in the past and many researchers have applied them for classification. A. Masnata and M. Sunseri [5] developed an automatic recognition system for weld defects. The system consists of a three-layered neural network where the input values

are a selection of the shape parameters obtained from the pulse-echo by Fischer analysis. Drai R et al. [6] extracted features for discrimination of detected echoes from the perspective of time domain, spectral domain and discrete wavelet representation. The compact feature vector obtained is then classified by different methods: the K nearest neighbor algorithm, the statistical Bayesian algorithm and artificial neural networks. Back-propagation neural networks are trained by the characteristic parameters extracted from ultrasonic signals to determine the type, location and length of cracks in the medium [7]. Cau et al. [8] developed a feed forward neural network along with wavelet blind separation to classify positions, width and depth of defects in not accessible pipes. Sahoo et al. [9] chose the peak amplitude, the energy of the signal and the time of the flight of ultrasonic echo signals as the feature indicators and developed a cascade feed forward back-propagation neural network model to estimate both crack size and crack location simultaneously. Two-dimensional information about different types of defects was collected via the wavelet transform and then used to improve automatic ultrasonic flaw detection and classification accuracies using the neural network [3]. Peng Yang et al. [10] presented automatic ultrasonic flaw signal classification models based on ANNs and support vector machines (SVMs) by using wavelet-transform-based strategies for feature extraction. Chen et al. [11] applied the wavelet packet transform to extract feature vectors representing defect qualities and then utilized that data to achieve the accurate classification of the welding defects an SVM-based radial basis function neural network.

Although the above-mentioned approaches have achieved good performance for automatic signal classification, it is necessary to choose and extract either time- or frequency-domain features from ultrasonic signals for training the neural network algorithm. However, this feature extraction is highly subjective and empirical, and in most cases, it is difficult to extract the reliable features that are most effective or relevant with the defects to be detected. In addition, it is not practical for industrial application since the workload of feature extraction for training neural networks increases dramatically under large amounts of original data. Therefore, it is very necessary to develop an automatic signal classification system which can automatically identify the most relevant features of ultrasonic signals with defects and eliminate the subjective selection.

Recently, because of the great strides in computing power, deep learning, which can extract and identify the effective features of images automatically, became a potential solution to deal with the above issue. In the field of deep learning, the convolutional neural network (CNN) has proved to be highly qualified for visual recognition tasks and is widely implemented [12]. A deep CNN with a linear SVM top layer was used to automatically extract features for each signal from wavelet coefficients and complete signal classification tasks for composite materials [13]. Guo et al. [14] converted laser ultrasonic signals into the scalograms (images) via the wavelet transform which were then used as the image input for the pretrained CNN to extract the defect features automatically to quantify the width of defects.

All above works have to utilize the wavelet transform technique to convert time-domain signals into images (scalogram) since traditional CNNs require high-resolution images as input. Such a process is still involved with the manual selection of parameters for wavelet transform and will be time-consuming with large volumes of raw data. In this paper, a CNN using raw time-domain data without the wavelet transform as input is designed to automatically extract features for the classification of ultrasonic signals. The approach is validated by the mockup with internal slots. The results demonstrate that our neural network architecture with independent features performs considerably better than the ones with handcrafted features. Finally, we visualize the shape, location and orientation of defects with a C-scan image based on classification results to validate the effectiveness of the results. Figure 1 shows the outline of the paper.

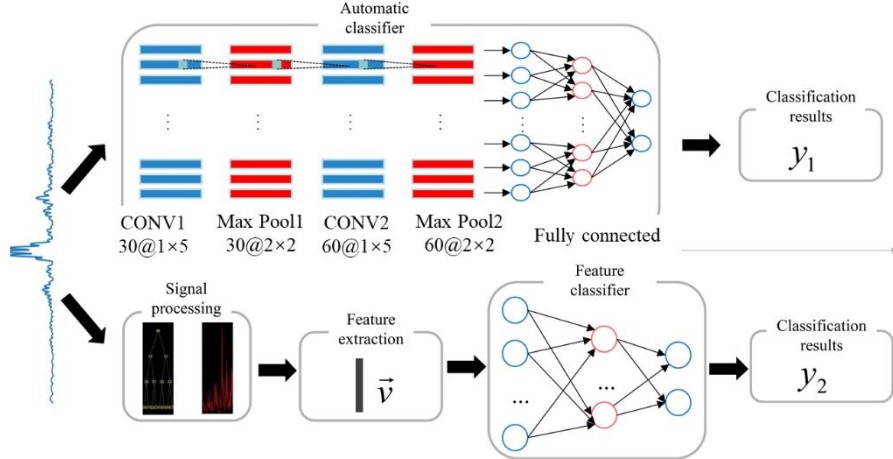

**Figure 1.** The workflow of our study on classifications of ultrasonic signals.

## 2. Experimental Setup

The sample used for the experiments is a circumferential weld composed of austenitic and martensitic stainless steel. The width and thickness of the weld were 10 mm and 13.6 mm, respectively, as shown in Figure 2a. Six slots were made by electrosparking every 60 degrees on the inner wall of the weld, as shown in Figure 2b. A 45-degree angle beam probe with 2 MHz center frequency was used to collect ultrasonic signals. The scanning area was within the range of 30 mm on both sides of the weld center line, as shown in Figure 2a. The probe stepped 1 mm along the direction perpendicular to the weld line (index axis) every scanning cycle till the end (scan axis) while the ultrasonic beam was parallel to circumferential direction.

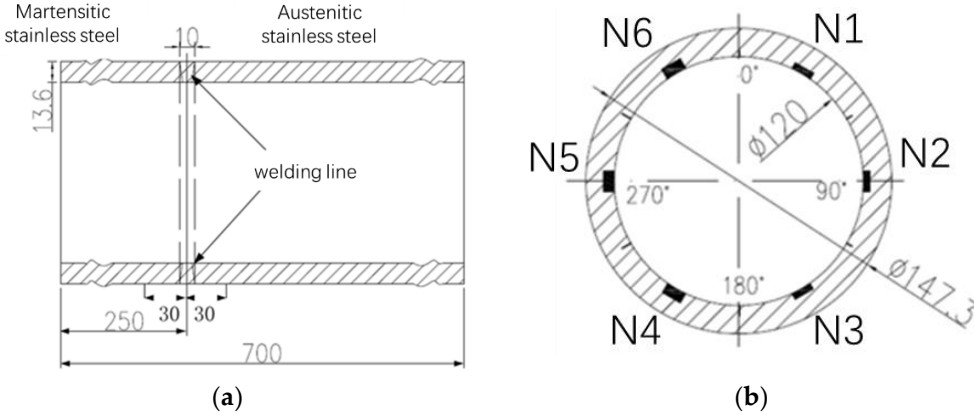

(a)

(b)

**Figure 2.** The detected circumferential weld: (**a**) the side view of the mockup; (**b**) the cross-section view of the mockup.

The ultrasonic detection experimental setup used for data collection is shown in Figure 3. It included three systems: a 35 MHz broadband of the ultrasonic transmitter-receiver system (UTRS) for generating and receiving the ultrasonic pulse from the transducers, a computer system based on a LabVIEW program for storing the ultrasonic data from detected specimens and a probe motion control system (PMCS) whose precision was 1 mm. Ultrasonic waves were transmitted and received using the same transducer excited by UTRS. The ultrasonic echoes collected from the detected specimens were then converted to electrical signals which were stored in the computer system next. The ultrasonic data acquisition card adopted NI PCI-5153 manufactured by National Instruments (NI). The card can receive an ultrasonic signal with sampling frequency of 2 GHz and resolution of 8 Bit.

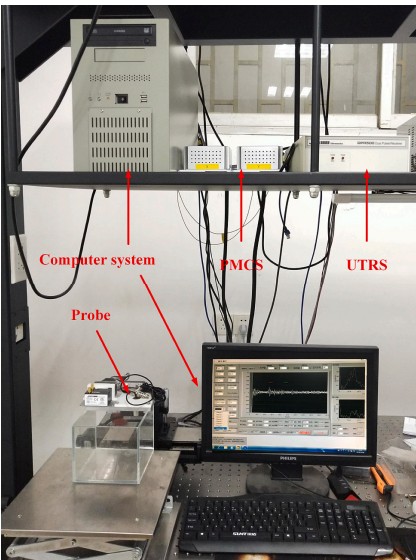

**Figure 3.** The ultrasonic detection experimental setup for data acquisition.

As a result, a database of 22,631 A-scan signals was acquired consisting of two classes: slot signals and nonslot signals. The database set was normalized between 0 and 1. Six thousand ultrasonic signals with equal numbers of slot signals and nonslot signals were statistically selected from the original database and randomly divided into two data sets called the training data set and testing data set. These two data sets were used to train and test neural networks where the number of training samples and testing samples was 5000 and 1000, respectively.

## 3. Method

The convolutional neural network is a type of deep neural network that has convolutional layers as well as fully connected layers. Convolution is a mathematical function that is widely used in the signal processing domain. In the convolutional neural network, convolutional layers actually utilize a cross-correlation technique that is technically very similar to convolution [15].

There are two important layers in CNNs, i.e., feature extraction layers and classification layers. Convolutional layers and pooling layers are feature-extracting layers, while classification layers are fully connected layers. Convolutional layers are not connected to every node in the input layer but to specific local regions based upon the defined filters/convolutional kernels. This architecture allows the network to concentrate on low-level features; these are then assembled into high-level features. CNN also has ability to learn a pattern at one location and then determine it in some other location. It is possible due to sharing the same parameter in filters. Pooling layer in the neural network is used to subsample the input that helps to reduce the computational load and also helps to avoid overfitting [15].

*Accuracy* was developed to evaluate the performance of the proposed CNN model on testing datasets. The accuracy of a dataset is defined as the number of correctly predicted observations divided by the total number of predictions made. Equation (1) gives the mathematical form of accuracy.

$$Accuracy = \frac{TP + TN}{TP + TN + FP + FN} \tag{1}$$

where *TP*, *TN*, *FP* and *FN* stand for true positive, true negative, false positive and false negative, respectively.

### 3.1. Architecture of the Proposed CNN

The convolutional neural network adopted in this paper was constructed using TEN-SORFLOW framework. The hardware platform for computation was a desktop with Intel core-i5 9400 CPU, 16 GB RAM, 256 GB SSD. Figure 4 shows the structure of the proposed CNN model made up of two convolution layers, two pooling layers, two fully connected layers and one output layer.

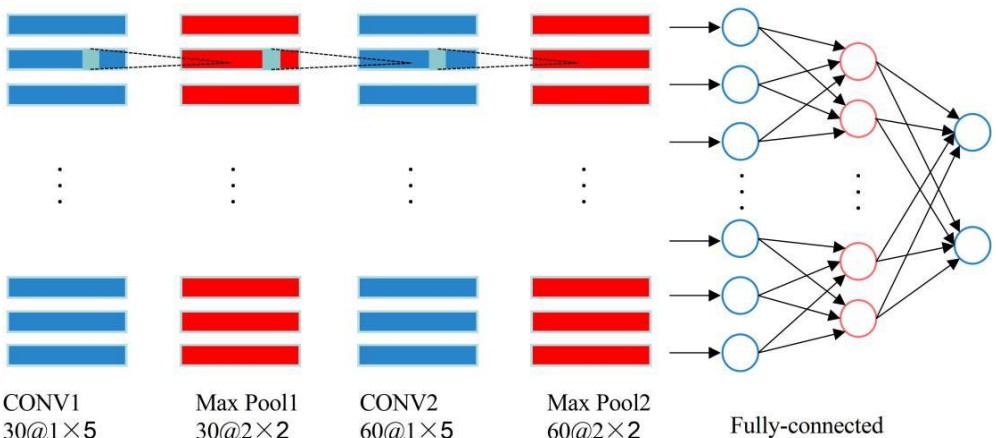

**Figure 4.** The structure of the proposed CNN.

The mean square error (*MSE*) was employed as loss function of the CNN, which is defined as

$$MSE = \frac{1}{N} \sum_{i=1}^{N} (\hat{y} - y)^2 \tag{2}$$

where $\hat{y}$ is the target value and $y$ is the prediction value of the network based on the input data. To enhance generalization and reduce the possibility of overfitting, regularized loss function $MSE_{reg}$ was used in training CNN after random initialization, which function is defined as

$$MSE_{reg} = \gamma MSE + \frac{1-\gamma}{n} \sum_{j=1}^{n} w_j^2 \tag{3}$$

where $w_j$ is the weight and $\gamma$ is a hyperparameter set manually.

In addition, an attenuation learning rate was adopted with initial value 0.01 to avoid gradient disappearance due to large learning rate. Rectified linear unit defined as $Relu(x) = \max(0, x)$ was applied to the activation function of the CNN. For making the CNN well robust, the moving average was applied for all training variables in the neural network.

### 3.2. Manually Selected Features

To demonstrate how much better the proposed method would perform than the handcrafted features method, a typical approach of handcrafted features was carried out for comparison.

In this approach, original signals were firstly processed by Fourier transform and wavelet transform to extract features manually in spectral domain and time-frequency domain, respectively. All extracted features were then used as input data to train a neural network. An eighteen-dimensional feature vector for each signal, $\vec{v}$, could be obtained as follows:

(1) Maximum amplitude in time domain: $Max_T = \max(T_i)$

(2) Minimum amplitude in time domain: $Min_T = \min(T_i)$

(3) Maximum difference in time domain: $\Delta_T = Max_T - Min_T$

(4)  Standard deviation in time domain: $Std_T = \sqrt{\frac{1}{n-1}\sum_{i=1}^{n}\left(T_i - \overline{T}\right)^2}$

(5)  Envelope area in time domain: $S_T = \sum_{i=1}^{n} T_i$

(6)  Maximum amplitude in spectral domain: $Max_F = \max(F_i)$

(7)  Minimum amplitude in spectral domain: $Min_F = \min(F_i)$

(8)  Maximum difference in spectral domain: $\Delta_F = Max_F - Min_F$

(9)  Standard deviation in spectral domain: $Std_F = \sqrt{\frac{1}{n-1}\sum_{i=1}^{n}\left(F_i - \overline{F}\right)^2}$

(10)  Envelope area in spectral domain: $S_T = \sum_{i=1}^{n} F_i$

where $T_i$ and $F_i$ are, respectively, amplitude values corresponding to $i$ sampling point in time domain and spectral domain and $\overline{T}$ and $\overline{F}$ are, respectively, mean amplitude values of signals in time domain and spectral domain. The other eight features are ratios of the energy of each frequency band to the total energy after wavelet decomposition of signals in the third layer.

## 4. Results and Discussion

### 4.1. Effect of Number of Middle Fully Connected Layer Neurons on Accuracy

The number of middle fully connected layer neurons is a key parameter that affects the ultimate accuracy of trained CNNs. It can also determine the number of features automatically extracted by CNNs. Obviously, changes in the number of these features will have significant impacts on the recognition rate of the proposed network. The performances of neural networks with different numbers of middle layer neurons were calculated using the testing data set as shown in Figure 5. It can be seen that the accuracy values of neural networks are all above 0.95 and shows a jagged rise eventually tending to the maximum of 0.972 where the initial number of middle-layer neurons is 500. Detailed information about accuracy variation with different numbers of middle-layer neurons is shown in Table 1.

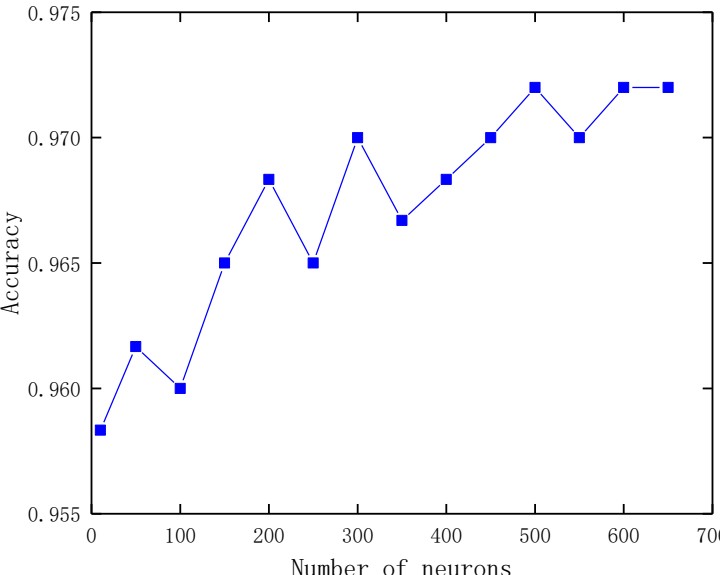

**Figure 5.** The accuracy variation of CNNs with the number of middle-layer neurons.

Based on the above research, a network structure with the configuration of 500 neurons in the middle fully connected layer was employed for signal identification in this task, which also implied that a 500-dimensional feature vector could characterize signals well enough from another perspective. Figure 6 shows the convergency curve of the loss function during the training of the network which could rapidly achieve convergency after only 80 epochs of 100 iterations.

**Table 1.** The accuracy variation with the number of middle-layer neurons.

| No. of the Middle Layer Neurons | Accuracy |
|---|---|
| 10 | 0.958 |
| 50 | 0.962 |
| 100 | 0.960 |
| 150 | 0.965 |
| 200 | 0.968 |
| 250 | 0.965 |
| 300 | 0.970 |
| 350 | 0.967 |
| 400 | 0.968 |
| 450 | 0.970 |
| 500 | 0.972 |
| 550 | 0.970 |
| 600 | 0.972 |
| 650 | 0.972 |

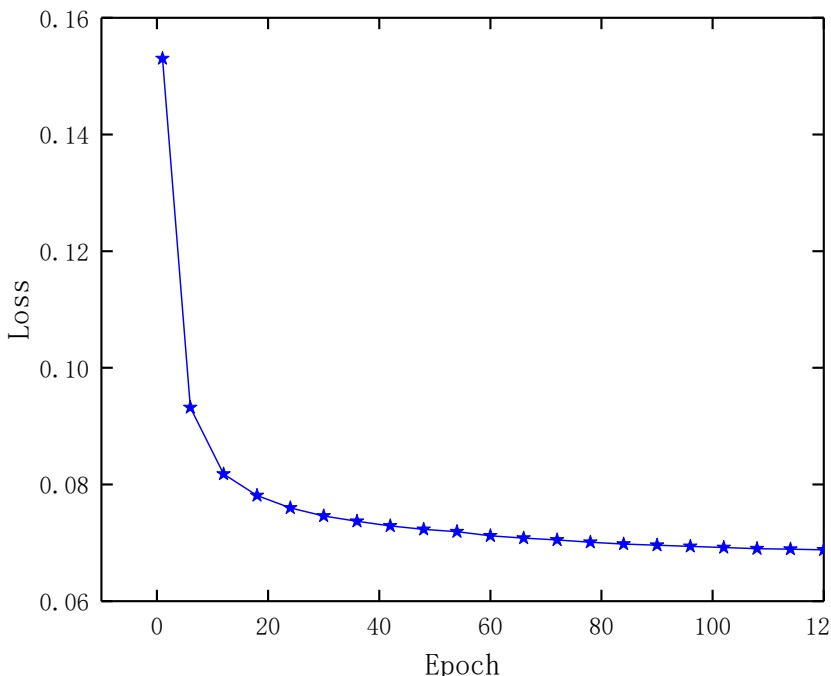

**Figure 6.** The convergency curve of loss function during training of the final determined CNNs.

*4.2. Comparison between the Proposed Approach and the Handcrafted Features Approach*

To validate the effectiveness of the proposed method, the handcrafted features approach was used for comparison. An eighteen-dimensional feature vector for each signal mentioned in Section 3.2 became an alternative, substituting original signals to train a classifier. The comparison results are shown in Table 2. The recognition rates in time domain, frequency domain, time-frequency domain and the fusion of the former three are listed respectively in the 'Handcrafted approach' column. It can be seen that testing accuracies in each individual domain are lower than the proposed method, but the fusion performance can slightly outperform the accuracy of the proposed method. However, the handcrafted features approach has to face the problem of severe workload burden with collected data increases. Balancing the efficiency and the accuracy, the proposed method is most suitable for industrial applications. It can be seen that accuracies in the manual extracting features method have a large fluctuation range from 0.796 to 0.981 by employing different features from Table 2. Such a problem can be avoided in the proposed method with good robustness.

**Table 2.** Performance comparison of the handcrafted features approach and the automatic approach.

| Methods | Handcrafted Approach | | | | Automatic Approach |
|---|---|---|---|---|---|
| | Time Domain | Frequency Domain | Time-Frequency Domain | Fusion | |
| Accuracy | 0.961 | 0.903 | 0.948 | 0.988 | 0.982 |

### 4.3. Visualization of Classification Results

The ultrasonic signals can be displayed in different ways for defect determination. Although A-scan is the common representation in nondestructive testing, it is usually beneficial to obtain a two-dimensional representation through a C-scan imaging process for defect determination. The C-scan shows the top view of the sample parallel to the scanning surface. By means of integrating amplitudes of A-scan signals within a time range over the surface, the C-scan image is able to confirm the presence of a defect in the sample according to the changes in amplitude. However, the traditional C-scan imaging process is too sensitive to the random noise during the scanning procedure.

As an alternative for the traditional C-scan imaging process, our C-scan image is produced according to the following steps:

- With the proposed classifier, the A-scan signal is indicated as a known type of defect classes which are provided as input for the C-scan imaging of the sample.
- For signal points in each defect class, a connected component labeling is performed on the C-scan image to further determine the defect area on the sample.
- We retain the points with neighbors belonging to the same class on the image and eliminate the others from the C-scan image as random noise.

Figure 7 is naive C-scan image generated based on the amplitude of original signals. There are a lot of noise signals in the whole weld area. It can be explained that the way to generate images seems too sensitive to the amplitude variation of original signals. The above-mentioned procedures are then used to form Figure 8, which consists of processed C-scan images based on the classification results of the (a) fusion method, (b) automatic method, (c) time domain method, (d) frequency domain method and (e) time-frequency domain method, respectively, corresponding to Table 2.

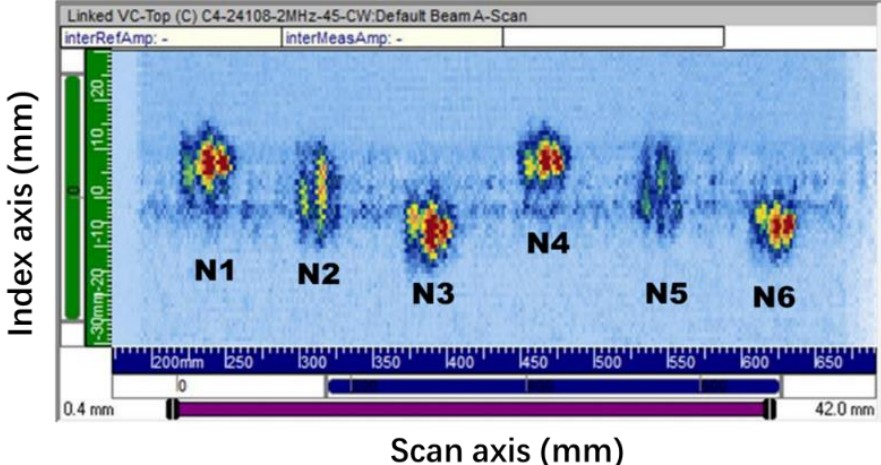

**Figure 7.** The C-scan image generated based on the amplitude of original A-scan signals obtained by detection.

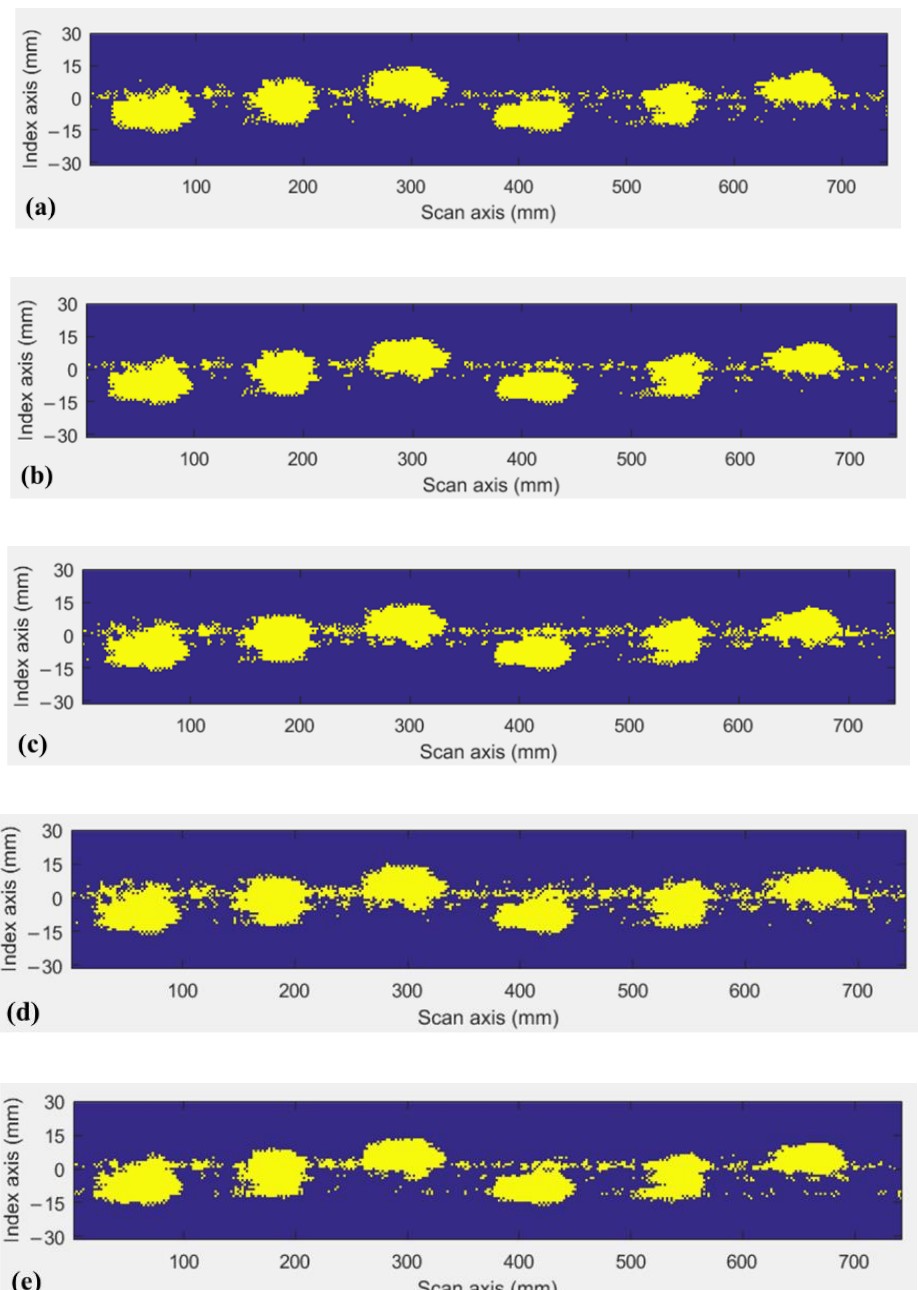

**Figure 8.** Processed C-scan images based on different classification results: (**a**) fusion; (**b**) automatic method; (**c**) time domain; (**d**) time-frequency domain; (**e**) frequency domain.

From Figure 8, some conclusions can be obtained as follows:

(1) The distribution of defects in each C-scan image conforms well to real defects made in the mockup.

(2) The image quality of defects gradually decreases from top to bottom in Figure 8, which is consistent with classification results.

(3) In terms of image quality, Figure 8a–e are superior to Figure 7. The proposed method is able to provide a higher credibility of C-scan visualization for defect determination.

(4) The proposed method still cannot completely remove noise signals and the next step is to develop effective denoising algorithms for more accurate defect identification.

## 5. Conclusions

Artificial intelligence, especially neural networks, has been increasingly utilized in ultrasonic signal classification. Conventional approaches necessarily require us to select and extract either time- or frequency-domain features from original ultrasonic signals manually for training the neural network model. However, this procedure is very subjective, empirical and time-consuming, and in most cases, it is very difficult to extract the appropriate features that are most relevant to the defects to be detected. In this paper, an automatic classification method using a convolutional neural network is proposed. The novelty of this work is to accomplish the automatic extraction of features from the original ultrasonic signals with raw time-domain data as input, avoiding the necessity and inaccuracy caused by manual feature selection.

An ultrasonic detection experimental setup was first used for data collection from the specimen with internal slots. Through the optimization of network structures, a network structure equipping the configuration of 500 neurons in fully connected network hidden layers was utilized for signal classification. To further validate the effectiveness of the proposed method, the handcrafted features either in the time- or frequency-domain approach were employed for comparison. The results demonstrate that the recognition rates in each individual domain are lower than the proposed method but the fusion performance can slightly outperform the accuracy of the proposed method. The proposed method shows good robustness and avoids the massive workload of manually extracting features. Finally, we visualized classification results to present and characterize detailed information about defects. Compared with the traditional C-scan based on amplitude of collected signals, our method is able to provide a higher credibility of C-scan visualization for defect determination.

The main objective of this work is to prove the feasibility and effectiveness of the proposed method, and therefore, only the specimen with internal slots is investigated. In real industrial applications, as long as sufficient data with regard to the various defect types (dimensions, orientations, locations and shapes) can be acquired, the developed method can achieve the accurate classification of corresponding defects.

**Author Contributions:** Conceptualization, Y.S.; Data curation, W.X.; Methodology, Y.S.; Validation, W.X.; Writing—original draft, Y.S.; Writing—review & editing, W.X., J.Z. and X.L. All authors have read and agreed to the published version of the manuscript.

**Funding:** This research was funded by the National Key R&D Program of China (Grant No. 2018YFB1106100).

**Data Availability Statement:** The data presented in this article are available on request from the corresponding author.

**Conflicts of Interest:** The authors declare no conflict of interest.

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
