# Peer review of "Automated Classification of Ultrasonic Signal via a Convolutional Neural Network"

_applsci, doi:10.3390/app12094179_

Round 1

Reviewer 1 Report

It is an interesting study but the manuscript  is poorly organized, and the presentation is neither clear nor concise. I would like to present the following comments and required changes to make this manuscript acceptable for publishing.

Language:

  1. The English in the manuscript is understandable but need serious improvement throughout the whole manuscript.
  2. Avoid using personnel pronouns throughout the manuscript.

Contents:

  1. I would strongly advise the authors to do a comprehensive literature review in the field towards the optimization of their research objectives. I think this is needed to be able to assess the improvement of measurement accuracy using your method compared to other states of the art methods.
  2. The introductory section should clearly explain the relevance of the presented research. It should also indicate, why the methodology used in the present study was chosen and why it will provide new insights.
  3. The font size of all figures should be increased throughout the paper. The font should be readable and understandable within the default view.
  4. Corroborate the results with the developed theory and algorithms.
  5. Conclusions must not be merely the repetition of the content of the preceding sections. The conclusion should concisely and clearly explain the significance and novelty of the results obtained in the presented work. Support your conclusion with the main quantitative data obtained in this research.

Author Response

Response to the comments from Reviewer 1

  1. I would strongly advise the authors to do a comprehensive literature review in the field towards the optimization of their research objectives. I think this is needed to be able to assess the improvement of measurement accuracy using your method compared to other states of the art methods.

Response: The introduction of this paper has been rewritten.

  1. The introductory section should clearly explain the relevance of the presented research. It should also indicate, why the methodology used in the present study was chosen and why it will provide new insights.

Response: The introduction of this paper has been rewritten.

  1. The font size of all figures should be increased throughout the paper. The font should be readable and understandable within the default view.

Response: All figures have been revised.

  1. Corroborate the results with the developed theory and algorithms.

Response: The results are corroborated and shown in Table 2.

  1. Conclusions must not be merely the repetition of the content of the preceding sections. The conclusion should concisely and clearly explain the significance and novelty of the results obtained in the presented work. Support your conclusion with the main quantitative data obtained in this research.

Response: The conclusions of this paper has been rewritten.

Reviewer 2 Report

Dear Authors,

It was a pleasure to review your paper. Find below some observations regarding it:

1. The novelty and the advantages of your method over others already cited in the literature are not sufficiently emphasized.

2. The experimental setup must me more detailed. How the ultrasonic signals were collected and processed, etc. A picture with it could much increase the credibility of the obtained results.

3. Generally, you should provide more details on the experiments and the classification itself. You should have in your mind when describing these, that the engineering community can be able to replicate (check) your work.

4. You should detail how the results given in Figs 6 and 7 were obtained.

5. You should also provide details concerning the way as the accuracies from Table 2 were obtained.

6. You should mandatorily mention the fields where the proposed method can be applied in the industry (besides that provided in the case study).

7. The conclusion should be not a simple overview of the paper or a reiteration of the given results. It should present the last word on the issues you raised in your paper, summarize your thoughts and convey the larger implications of your study, demonstrate the importance of your ideas, and introduce possible new or expanded ways of thinking about the research problem in the discussion.

8. The quality of the figures is very poor.

9. The references should be edited upon the given template! Generally, you should follow more precisely the formatting requirements (see equation cations, etc.).

Author Response

Response to the comments from Reviewer 2

  1. The novelty and the advantages of your method over others already cited in the literature are not sufficiently emphasized.

Response: The introduction of this paper has been rewritten.

  1. The experimental setup must me more detailed. How the ultrasonic signals were collected and processed, etc. A picture with it could much increase the credibility of the obtained results.

Response: The experimental setup is shown in Figure 3.

  1. Generally, you should provide more details on the experiments and the classification itself. You should have in your mind when describing these, that the engineering community can be able to replicate (check) your work.

Response: Please see page 3, line 111~120 of the revised version.

  1. You should detail how the results given in Figs 6 and 7 were obtained.

Response: After the revision, Figs 6 and 7 become Figs 8 and 7. Figure 8 was obtained as described in page 8, line 252-259 of the revised version. Figure 7 was obtained as described in page 8, line 244-250 of the revised version.

  1. You should also provide details concerning the way as the accuracies from Table 2 were obtained.

Response: Details concerning the way as the accuracies from Table 2 were described from line 150 to 156 of the revised version.

  1. You should mandatorily mention the fields where the proposed method can be applied in the industry (besides that provided in the case study).

Response: Please see 10, line 311-315 of the revised version.

  1. The conclusion should be not a simple overview of the paper or a reiteration of the given results. It should present the last word on the issues you raised in your paper, summarize your thoughts and convey the larger implications of your study, demonstrate the importance of your ideas, and introduce possible new or expanded ways of thinking about the research problem in the discussion.

Response: The conclusion of this paper has been rewritten.

  1. The quality of the figures is very poor.

All figures have been revised.

  1. The references should be edited upon the given template! Generally, you should follow more precisely the formatting requirements (see equation cations, etc.).

All references have been revised.

Round 2

Reviewer 1 Report

The authors have well revised and enhanced the quality of the article. I am satisfied with the revisions made and the manuscript could be considered for publication.

Reviewer 2 Report

Thank you for considering all my observations regarding your paper.